# Continuously Reinforced Carbon Nanotube Film Sea-Cucumber-like Polyaniline Nanocomposites for Flexible Self-Supporting Energy-Storage Electrode Materials

**DOI:** 10.3390/nano12010008

**Published:** 2021-12-21

**Authors:** Bingjian Li, Shi Liu, Haicun Yang, Xixi Xu, Yinjie Zhou, Rong Yang, Yun Zhang, Jinchun Li

**Affiliations:** 1School of Materials Science and Engineering, Changzhou University, Changzhou 213164, China; lbj7151@163.com (B.L.); liushi19890101@126.com (S.L.); yhcbobo@cczu.edu.cn (H.Y.); xxx970209@163.com (X.X.); zhouyinjie41@163.com (Y.Z.); cloudyyang@cczu.edu.cn (R.Y.); 2Jiangsu Key Laboratory of Environmentally Friendly Polymeric Materials, Changzhou University, Changzhou 213164, China; 3Changzhou Key Laboratory of Functional Film Materials, Changzhou 213164, China; hannah0422@126.com; 4National-Local Joint Engineering Research Center of Biomass Refining and High-Quality Utilization, Changzhou 213164, China

**Keywords:** carbon nanotube film, polyaniline, cyclic voltammetry electrochemical polymerization, flexible self-supporting, sea-cucumber-like 3D nanoprotrusion structure

## Abstract

The charge storage mechanism and capacity of supercapacitors completely depend on the electrochemical and mechanical properties of electrode materials. Herein, continuously reinforced carbon nanotube film (CNTF), as the flexible support layer and the conductive skeleton, was prepared via the floating catalytic chemical vapor deposition (FCCVD) method. Furthermore, a series of novel flexible self-supporting CNTF/polyaniline (PANI) nanocomposite electrode materials were prepared by cyclic voltammetry electrochemical polymerization (CVEP), with aniline and mixed-acid-treated CNTF film. By controlling the different polymerization cycles, it was found that the growth model, morphology, apparent color, and loading amount of the PANI on the CNTF surface were different. The CNTF/PANI-15C composite electrode, prepared by 15 cycles of electrochemical polymerization, has a unique surface, with a “sea-cucumber-like” 3D nanoprotrusion structure and microporous channels formed via the stacking of the PANI nanowires. A CNTF/PANI-15C flexible electrode exhibited the highest specific capacitance, 903.6 F/g, and the highest energy density, 45.2 Wh/kg, at the current density of 1 A/g and the voltage window of 0 to 0.6 V. It could maintain 73.9% of the initial value at a high current density of 10 A/g. The excellent electrochemical cycle and structural stabilities were confirmed on the condition of the higher capacitance retention of 95.1% after 2000 cycles of galvanostatic charge/discharge, and on the almost unchanged electrochemical performances after 500 cycles of bending. The tensile strength of the composite electrode was 124.5 MPa, and the elongation at break was 18.9%.

## 1. Introduction

The current rapid development of lightweight, flexible, and wearable electronic devices, such as smart glasses, elegant watches, bracelets, flexible folding-screen mobile phones, and biosensors, etc., involve the fields of wearable devices, implantable equipment, biomedical treatment, and others [1,2,3,4,5,6]. The flexible super energy storage has a bright prospect because of the continuously increasing demand for portable devices and wearable electronic products [7,8,9,10,11,12].

Carbon nanotubes (CNTs) are a kind of nanomaterial that has an intrinsic high-aspect ratio, electrical conductivity, chemical stability, and an extraordinary mechanical strength and modulus, which endow them with the capacity to prepare nanocomposites with excellent electronic and mechanical properties. Up until now, tremendous efforts have been made to prepare the conductive composites of CNTs, in which polyaniline/CNT composites (PANI/CNTs) have demonstrated multiple applications in thermoelectric devices, artificial muscles, sensors, and capacitors. For instance, He et al. prepared an innovative “sea-cucumber-like” PANI/CNTs-COOH composite material, which exhibited a higher specific capacitance and longer-term cycling stability than the PANI/CNTs [13]. The novel “sea-cucumber-like” material, with a three-dimensional (3D) structure, provided a high electrode/electrolyte contact area, and a short path length for electrons and electrolyte ion transport. Hussain et al. covered the vertically aligned CNTs with the PANI thin film, applying RF-plasma polymerization, and this composite showed a notably higher specific capacitance and cycling stability because of the special rank of the CNTs [14]. Therefore, because of the higher capacitance of PANI, and the higher conductivity and mechanical strength of CNTs, the combination of PANI and CNTs has been confirmed as an effective avenue for obtaining nanocomposites with remarkably enhanced electrochemical and mechanical properties [15]. In addition, other carbon materials, such as carbon nanofibers, graphene, and carbon nanospheres, have been used to enhance the comprehensive performances of nanocomposite electrodes through the synergistic effect between PANI and these carbon nanomaterials [16,17,18]. The cyclic stability, especially, can be significantly improved with the introduction of conductive nanomaterials.

However, the poor processability of conductive polymers, and the aggregation of the inorganic nanomaterials make it difficult to control the microstructures of the composite electrodes, limiting their application because of the relatively brittle defect [19]. Hence, to solve this defect, the structure of the traditional flexible supercapacitor always includes the current collector (stainless steel fabric or foamed nickel) and the binder (polytetrafluoroethylene or polyvinylidene fluoride). Because of the presence of additional current collectors and nonconductive polymer binders, redundant contact resistance and a useless quality will be generated, which will seriously affect the performance of the flexible capacitors [20,21]. Self-supporting conductive polymer-based nanocomposite films have been confirmed as binder-free electrodes with enhanced pseudocapacitive performances. Self-supporting composite electrodes, such as carbon nanotube paper (CNTP), carbon nanofiber film (CNFF), graphene hydrogel (GOH), and carbon monolith (CM), have been investigated, and the preparation method involves the layer-by-layer assembly, filtration, and solution casting methods [22,23,24,25,26,27]. Nevertheless, most of the above composites are not flexible and robust enough during mechanical bending. They crack more easily under bending force without a supporter, and the intrinsic porous structures usually tend to collapse because of the weaker interconnections. Because of the atom-thick 2D carbon nanostructure, constructed by the strong π-π interactions, graphene is easier to transform into a graphite-like structure without a high specific surface area, a sufficient electrically porous channel, and a faradaic reaction [28]. It is still a challenge to easily prepare flexible CNT-based composite electrodes with excellent mechanical properties and electrochemical performances, and practical application in supercapacitors.

CNTF usually exhibits the typical electrical double-layer capacitors (EDLC), with the advantages of a large specific surface area and high power density. However, the disadvantages, such as the relatively lower energy density and specific capacitance, also need to be overcome by increasing the porosities and specific surface areas of carbon materials [29,30]. On the other hand, PANI presents a typical pseudocapacitor and a high specific capacitance value, but a lower power density, terrible cycle stability, and a poor rate performance [31]. Growing the PANI layer from the surface of the CNTF, and controlling the micromorphology of PANI, are effective solutions to overcoming the drawbacks of CNTF and PANI, respectively [32,33]. Replacing the current collector and the adhesive in the traditional supercapacitor structure with CNTF can avoid the generation of redundant contact resistance and useless quality. The flexible self-supporting material (e.g., CNTF) of the conductive skeleton shortens the electron transmission distance and heightens the volume ratio of the interconnected gap in the electrode. In addition, CNT-based flexible self-supporting materials are light and have excellent flexibility and high conductivity [34,35,36]. Because of the flexible conductive frameworks, preparing CNTF-based composites as a new flexible self-supporting material seems to be a more practical approach, storing the charge through the coordination of faradaic and nonfaradaic reactions [37,38,39,40].

Inspired by the above thoughts, in this work, the continuously enhanced CNTF was prepared by using the FCCVD method and was then modified via the mixed acid treatment. Subsequently, the PANI conductive polymer layer was grown on the surface of the acidified CNTF via the CVEP method to prepare flexible self-supporting nanocomposite electrode materials with controllable microstructures and great comprehensive performances. Finally, the influence of the microstructure of the PANI at different polymerization cycles on the electrochemical performance was investigated in detail.

## 2. Materials and Methods

### 2.1. Materials

The aniline (ANI) monomer was purchased from the Sinopharm Chemical Reagent Co., Ltd. (≥99.5%, Shanghai, China), and was purified by vacuum distillation before use. Concentrated sulfuric acid (H_2_SO_4_, 98%) was obtained from the Sinopharm Chemical Reagent Co., Ltd (Shanghai, China). Concentrated nitric acid (HNO_3_, 68%) was supplied by the Sinopharm Chemical Reagent Co., Ltd (Shanghai, China). Other reagents were analytical reagent grade and were used directly.

### 2.2. Preparation of CNTF

The continuously reinforced CNTF with excellent flexibility was prepared by the FCCVD method by using absolute ethanol as the carbon source precursor (95 wt%), ferrocene as the catalyst (2 wt%), and thiophene as the accelerator (3 wt%). Typically, the mixture of quantitative ethanol, ferrocene, and thiophene was firstly ultrasonicated, and then injected into the low-temperature zone (400 °C) of a vertical high-temperature cracking furnace through a syringe, at a constant rate (0.15 mL/min). Hydrogen and argon, with a volume ratio of 3:1, were used as the carrier gas, and the flow rate was 2200 mL/min. Subsequently, the carbon source was gasified and blown by the carrier gas to the high-temperature zone (1350 °C). Finally, the CNTs were grown gradually, and were floated to the furnace body mold with the driving of the carrier gas. They were then collected on a roller collector after water infiltration. With the axial rotation and horizontal moving of the roller collector, a continuously reinforced CNTF, with a uniform thickness of 15.0 μm, was successfully prepared by layer-by-layer compression.

### 2.3. Preparation of Flexible Self-Supporting CNTF/PANI Nanocomposite Films

The CNTF, with a size of 10 mm × 20 mm, was immersed in an ethanol/acetone (1:1) mixture and was ultrasonicated for 2 h to clean the surface of the CNTF, and was then washed with deionized water and dried in a vacuum oven at 80 °C for 4 h. The cleaned CNTF was immersed in the mixed acid (H_2_SO_4_/HNO_3_ = 3:1) for 150 min at room temperature, and was then washed with deionized water and dried in a vacuum oven at 80 °C for 4 h. The acidified CNTF was obtained. A series of CNTF/PANI nanocomposite films were prepared by the CVEP method, in the potential range from −0.2–0.8 V vs. the saturated calomel electrode (SCE) at a sweep rate of 10 mV/s, with a platinum foil and an SCE as the counter electrode and reference electrode, respectively. The acidified CNTF was immersed into the mixed electrolyte, consisting of 1.0 M H_2_SO_4_ and 0.2 M ANI monomer. ANI monomer was oxidized to ANI cationic radicals and polymerized on the surface of acidified CNTF to form a PANI layer. The series of nanocomposite films, noted as CNTF/PANI-1C, CNTF/PANI-3C, CNTF/PANI-5C, CNTF/PANI-10C, and CNTF/PANI-15C, were prepared by different scanning cycles (1 C, 3 C, 5 C, 10 C, and 15 C). The abovementioned prepared nanocomposite films were washed with deionized water until the pH of the washing water reached 7.0, and they were then dried at 80 °C in a vacuum oven for 12 h. The loading amount of the PANI was determined by calculating the changes of the mass and thickness of the CNTF before and after electrochemical polymerization.

### 2.4. Characterization

Transmission electron microscopy micrographs (TEM) were conducted on a JEM-1200 EX/S TEM, with an accelerating voltage of 200 kV (JEOL, Akishima, Tokyo, Japan). Particles samples for the TEM were prepared by placing 10-μL particle dispersions (0.5 g/L in chloroform) on copper grids coated with a perforated carbon film. Field emission scanning electron microscopy (FESEM) micrographs were carried out on a SUPRA-55 FESEM, with an accelerating voltage of 5 kV (Zeiss, Oberkochen, Baden-Wuberg, Germany). Fourier transform infrared (FTIR) spectra were recorded on an Avatar 460 FTIR spectrometer (Thermo Nicolet, Waltham, MA, USA), in the frequency range of 500–4000 cm^−1^, by using the KBr pellet method and the transmission mode. The Raman spectra were recorded on a DXR laser confocal Raman spectrometer (Thermo Scientific, Waltham, MA, USA). The crystal structures of the electrodes were analyzed by X-ray diffraction (XRD, Shimadzu, Japan). The surface compositions were investigated by X-ray photoelectron spectroscopy spectra, collected on an Escalab 250Xi spectrometer (Thermo Scientific, Waltham, MA, USA), with a nonmonochromatic X-ray source (Al Kα radiation as the exciting source). The tensile strength was measured using a microcomputer-controlled electrical universal material testing machine (MTS Systems Corporation, Eden Prairie, MN, USA), with a 5-kN load cell and a 25-mm/min testing speed. The thicknesses of the samples were measured with a micrometer (Model CH-1-S, Shanghai Liuling Instrument Factory, Shanghai, China). The data were the average of at least five measurements.

### 2.5. Electrochemical Measurements

The electrochemical performance of the nanocomposite electrodes was studied through cyclic voltammetry (CV), galvanostatic charging–discharging (GCD), and electrochemical impedance spectrum (EIS) measurements, by using an electrochemical workstation (CHI660E, Shanghai, China). In the three-electrode system, the PANI/CNTF nanocomposite electrodes (10 × 5 mm) were used as the working electrodes, without any additional current collector, and the platinum foil and saturated calomel electrode (SCE) were used as the counter electrode and the reference electrode, respectively, in the electrolyte of the 1.0-M sulfuric acid aqueous solution. The CV curves were recorded with a CV test potential window from −0.2 to 0.8 V, under different scanning speeds (5, 10, 20, 50, and 100 mV/s). The GCD curves were recorded with a GCD test potential window of 0 to 0.6 V, under different current densities (1, 2, 5, and 10 A/g). The EIS curves were recorded in the scanning test frequency range from 0.01 Hz to 100 kHz, with an AC-voltage amplitude of 5 mV. The specific capacitance of the nanocomposite electrodes can be calculated through the following, Equation (1), based on the GCD results:(1)$$C=\left(I\times \Delta t\right)/\left(\Delta V\times m\right)$$
where *C* is the specific capacitance (F/g); *I* is the discharge current (A) during the GCD test; Δ*t* is the discharge time; Δ*V* is the potential change during the discharge; and *m* is the weight of the nanocomposite electrode. The energy density (*E*) and the power density (*P*) of the nanocomposite electrode were calculated via Equations (2) and (3): (2)$$E=0.5\times C\times \Delta V^{2}$$
(3)P=E/Δt
where *C* is the specific capacitance (F/g); Δ*V* is the operating voltage window; *P* is the power density; and Δ*t* is the discharge time.

## 3. Results and Discussion 

### 3.1. Structural, Morphological, and Textual Analyses

The typical CV curves of the nanocomposite films prepared via the CVEP method after 5 C, 10 C, and 15 C, at a scan rate of 10 mV/s, are shown in Figure 1. The appearances of the obvious oxidation and reduction reaction peaks, and the reduction reaction peaks of the aniline cations during each cycle, were in line with the results reported previously [41,42]. The humps (A_1_, A_2_, and A_3_) within the CV curves resemble the polymeric oxidation reactions, and the dips (C_1_, C_2_, and C_3_) resemble the reduction reactions of the ANI monomer. The expansion curves of each layer in the CV curves represent that both the thickness and the load amount of the PANI layer increase with the increase in the number of cycles (Table 1). The thickness of the pure CNTF was 15 μm, and the mixed acid treatment can promote the growth of PANI from the surface of CNT fibers and increase the interface adhesion. The thicknesses of the PANI layers were increased to 22.6 and 26.3 μm, and the loading amounts of the PANI were 62.5 wt% and 73.2 wt%, respectively, after CVEP for 1 time, and CVEP for 3 times. Meanwhile, the color of the CNTF/PANI nanocomposite film was gradually changed to purple (Appendix A). The loading amount and thickness of the PANI were increased with the increase in the cycle time, and reached 121.5 wt% and 30.7 μm, respectively, with a gradually apparent A_2_/C_2_ peak in the CV curves, and a reddish-brown (Appendix A) color of the film after five cycles. After the tenth cycle of polymerization, the loading of the PANI reached 163.1 wt%, the thickness of the CNTF/PANI-10C increased to 51.1 μm, and the nanocomposite showed a blue-green color (Appendix A). Finally, the loading amount and the thickness of the PANI reached 441.1 wt% and 143.1 μm, respectively. The nanocomposite film with a dark-green color (Appendix A) was obtained after the fifteenth cycle of polymerization. The above results show that the ANI monomer can be oxidized into cationic free radicals by the cyclic voltammetry electrochemical oxidation-reduction method, and then coupled to form a uniform PANI polymer layer on the surface of the CNTF. With the periodic redox reaction, the color of the PANI layer on the surface of the CNTF changed repeatedly and presented the corresponding color after the end of each cycle because of the changed structure of the PANI. During the electrochemical polymerization process, the ratio of benzene (reduction unit) to quinone (oxidation unit) in the molecular chain of the PANI in different cycles changed, representing the different degrees of the redox state for the PANI.

The surface morphologies of the pure CNTF, CNTF/PANI-1C, CNTF/PANI-3C, CNTF/PANI-5C, CNTF/PANI-10C, and CNTF/PANI-15C are shown in Figure 2. It can be seen from Figure 2A,B that the pure CNTF prepared by the FCCVD method presented crisscrossing CNT fibers and fiber bundles, composed of 5–8 single CNT fibers, and that the diameter of a single CNT fiber was 20–30 nm. The CNT fibers and fiber bundles were intertwined to form a porous skeleton with excellent flexibility, and there were no noticeable changes in the surface structures after the mixed acid treatment. The fiber diameter reached 50–70 nm and 80–100 nm, after 1 cycle of CVEP and 3 cycles of CVEP, respectively (Figure 2C–F), which indicates that the PANI was polymerized and grown uniformly, with the acidified CNT fibers as the coaxial center. The reactive functional group (-COOH, -OH, and -NH_2_) between the acidified CNT fibers and the monomer promoted the polymerization of the ANI from the surface of the CNT fibers. After the fifth cycle of CVEP, the CNT fiber network structure in the CNTF/PANI-5C was produced, and it can be observed in Figure 2G,H. The PANI layer grown from the single CNT surface densely covered the gaps between the original CNT fibers and the fiber bundles, which can improve the interfacial bonding between the CNT fiber and the conductive PANI, reduce the contact resistance, and enhance the overall conductivity [43]. Interestingly, a large number of 3D nanoprotrusions on the surface of the PANI layer were formed and developed into a nanowire structure, increasing the specific surface area of the PANI layer (marked in Figure 2G).

Subsequently, the structure of the PANI nanowires on the surface of the CNTF/PANI-10C was further developed to the “sea-cucumber-like” structure, with a diameter of 100–200 nm, and a larger aspect ratio. It should be noted that the PANI nanowires resulting from the originally ordered 3D nanoprotrusion structure of the CNTF/PANI-5C, and the “sea-cucumber-like” structure wrapped in the inner CNTF fiber will further develop into the new nanowire structure. Accordingly, another new “sea-cucumber-like” skin, with a 3D nanoprotrusion structure, will be continuously established from the newly generated nanowire, again and again (Figure 2J and Figure 3). The uniform and ordered PANI nano “sea-cucumber-like” structure, formed on the basis of the above growth rule, increased the specific surface area of the PANI layer, and provided enough active sites for ion adsorption (Figure 2K–L and Figure 4D). Consequently, a three-dimensional porous microstructure was formed by the compact stack of the plentiful nano “sea-cucumber-like” structures without intrinsic micropores. The specific surface area that the Brunauer–Emmett–Teller (BET) analysis reached was 39 m^2^/g, which is beneficial to the improvement of the electrical properties of the CNTF/PANI as a supercapacitor flexible self-supporting electrode material. It can also be seen from the TEM images that the CNT’s fiber core was evenly wrapped by the PANI energy-storage layer, and that a dendritic structure, with a diameter of about 200 nm, began to appear in the CNTF/PANI-5C. In addition, the “sea-cucumber-like” 3D nanoprotruding structures in the PANI CNTF/PANI-10C and CNTF/PANI-15C were also observed (Figure 4C,D), which was consistent with the FESEM results.

The acidified CNTF, CNTF/PANI-1C, CNTF/PANI-3C, CNTF/PANI-5C, CNTF/PANI-10C, and CNTF/PANI-15C nanocomposite films were confirmed by FTIR. As shown in Figure 5A, the acidified CNTF showed characteristic absorption peaks of C=O and O-H at 1730 and 1634 cm^−1^, respectively, indicating that the CNT fiber surface was acidified by the mixed acid [44]. For the CNTF/PANI, the peaks at 1585 and 1494 cm^−1^ were attributed to the C=C stretching vibration of the quinone ring of the PANI structure [45,46]. The peak at 1299 cm^−1^ was assigned to the stretching vibration of the C=N-C (the quinone ring) of the PANI, and the peak at 1243 cm^−1^ corresponded to the asymmetric -NH_2_^+^- stretching vibration in the polaron structure [47]. The vibration of protonated quinoid imine =NH^+^- in the doped PANI molecular chain was found at 1121 cm^−1^, and the peak at 821 cm^−1^ corresponded to the C–H out-of-plane bending vibration on the disubstituted benzene ring, and the broad absorption peak corresponded to the stretching vibration peak of the N–H bond in the PANI structure, which appeared at 3464 cm^−1^. All the above results indicate that the ANI monomer was successfully electrochemically polymerized and deposited on the surface of the acidified CNTF. However, with the increase in the polymerization cycle period, the characteristic peak shifted to the low wavenumber and gradually became broad and strong. Compared with the first polymerization cycle, the peaks of C=C (quinone ring), C=N-C, -NH_2_^+^-, =NH^+^-, and C-H of CNTF/PANI-15C were all located at the lower wavenumbers of 1560, 1490, 1297, 1241, 1105, and 801 cm^−1^, respectively. This was mainly due to the increase in the electron delocalization in the PANI molecular chain during the protonation process, which formed a conjugated structure and reduced the vibration frequency of each group. The shifting of the characterization peaks toward the low-wavenumber direction suggests the highest protonation degree of the CNTF/PANI-15C, with a highly conductive form called the "emeraldine" state of salt [48].

In addition, the C=C peaks of the quinone ring in the CNTF/PANI-1C and CNTF/PANI-3C were stronger than those of the benzene ring, indicating the higher oxidation degree of the PANI. However, the above intensity contrast was reversed, and the characteristic peaks of C-N, -NH_2_^+^-, and =NH^+^- were gradually increased after five cycles of polymerization. After the doping of sulfuric acid, the nitrogen atoms in the quinone structure of the PANI can interact with sulfuric acid to generate polarons, and the delocalization of these polarons promotes the transformation from the quinone ring to the benzene ring. The above results indicate that the microstructures of the PANI macromolecules were changed during the electrochemical polymerization, which corresponds to the analysis in the FESEM and TEM.

The Raman spectra of the pure CNTF, acidified CNTF, CNTF/PANI-1C, CNTF/PANI-3C, CNTF/PANI-5C, CNTF/PANI-10C, and CNTF/PANI-15C are shown in Appendix A. The pure CNTF presented three peaks, at 1338, 1571, and 2677 cm^−1^. The D band at 1338 cm^−1^ indicated the existence of the sp^3^ hybridization of the carbon atoms, attributed to the amorphous carbon and the defects of the CNTs. The G band at 1571 cm^−1^ was related to the in-plane bond stretching motion of the C sp^2^ atom pair, reflecting the regular graphitization degree in the CNTF [49,50]. The peak at 2677 cm^−1^ was assigned to the G’ band that originated from the 2D peak of a small amount of graphene in the CNTF [51,52]. The I_D_/I_G_ area ratio was 0.18, indicating that pure CNTF has a high degree of graphitization. The I_D_/I_G_ area ratio of the CNTF film increased to 0.88 after the mixed acid treatment, indicating that part of the graphitized structure was destroyed during the treatment. As for the spectra of the CNTF/PANI, the peak at 1585 cm^−1^ represented the C=C stretching vibration of the quinone ring, and the peak at 1468 cm^−1^ corresponded to the semiquinone radical cation structure in the PANI. The C–H and C–N stretching vibrations were found at the peaks of 1162 and 1214 cm^−1^, respectively. The C=C stretching vibration peaks of the CNTF/PANI-10C and CNTF/PANI-15C were merged with the G bands of the CNTs, generating a broad peak at 1571 cm^−1^. The peaks at 413 and 513 cm^−1^ were derived from the vibration peaks of -NH_2_. All the abovementioned peaks confirm that the PANI was successfully electrochemically polymerized on the surface of the CNTF. With the increase in the electrochemical polymerization period, the intensities of the peaks at 1162, 1214, and 1468 cm^−1^ were gradually increased. Meanwhile, the gradual increase in the I_D_/I_G_ area ratio, and the disappearance of the 2D characteristics of the graphene at 2677 cm^−1^ indicate the increase in the content of the sp^3^ defect and the gradual thickening of the PANI layer. After the fifth cycle of cyclic voltammetry electrochemical polymerization, all the characteristic peaks of the PANI in the CNTF/PANI-10C and CNTF/PANI-15C became stronger. Importantly, the continuously strengthening peak at 1380 cm^−1^, corresponding to the -NH_2_^+^- of the semiquinone ring, indicate that the protonation degree of the PANI macromolecular chains gradually increased.

Figure 5B shows the XRD patterns of the pure CNTF, acidified CNTF, CNTF/PANI-1C, CNTF/PANI-3C, CNTF/PANI-5C, CNTF/PANI-10C, and CNTF/PANI-15C. The diffraction peak at 2θ = 26.0° of the CNTF corresponded to the (002) crystal plane [44,53]. All the CNTF/PANI show three diffraction peaks, at 2θ = 15.2°, 22.0°, and 25.8°, corresponding to the (011), (020), and (200) crystal planes of the PANI, respectively [32,54,55,56]. With the increase in the electrochemical polymerization period, the main reflection positions in the XRD pattern of the CNTF/PANI were exactly the same, except for the visible change in the reflection intensity, which indicates that the crystal form of the PANI remained unchanged, even though there was the transform in the configuration of the PANI main chain after doping and protonation.

The XPS analysis was used to identify the chemical compositions and electronic structures of the acidified CNTF and CNTF/PANI nanocomposite electrodes. As shown in Figure 6A, the acidified CNTF showed two peaks, at 285.1 eV and 532.1 eV, corresponding to the C 1s and O 1s. The CNTF/PANI presented an obvious peak at 400.1 eV of N 1s, coming from the PANI after electrochemical polymerization. It is worth noting that the O 1s peak of the CNTF/PANI was higher than that of the acidified CNTF, which is mainly ascribed to the delocalization of the polarons formed by the SO_4_^2−^ in the electrolyte and the protonated nitrogen atom in the PANI. In Figure 6B, the C 1s signal of CNTF/PANI-15C can be curve-fitted with four peaks, at 284.3 eV, 285.5 eV, 286.6 eV, and 288.3 eV, corresponding to C–C/C=C, C–N, C–O, and C=O, respectively [57,58]. In Figure 6C, the N 1s signal of the CNTF/PANI-15C can also be curve-fitted with four peaks, at 398.6 eV, 399.3 eV, 400.7 eV, and 402.2 eV, attributed to the =N- of the quinoid imine unit, the -NH- of the benzenoid amine unit, the =NH^+^- of the protonated imine unit, and the -NH_2_^+^- of the protonated amine unit, respectively [59]. The area ratio of S_1_ (=N-):S_2_ (-NH-):S_3_ (=NH^+^-):S_4_ (-NH_2_^+^-) was calculated to be 0.04:0.51:0.31:0.14, indicating that the doping rate of the CNTF/PANI-15C was attained to 45%. In addition, by comparing the content of benzenoid amine -NH-, it was found that the =N- of the quinoid imine unit was significantly reduced with the increase in the =NH^+^- of the protonated imine unit. It can be concluded that the protonation reaction during the doping process was mainly on the nitrogen atom of the quinoid imine.

The stress–strain curves of the acidified CNTF, CNTF/PANI-5C, CNTF/PANI-10C, and CNTF/PANI-15C are shown in Figure 7. It can be seen that the tensile strength of the acidified CNTF was 111.8 MPa, and that the elongation at break was 20.2%. After five polymerization cycles of cyclic voltammetry, the tensile strength of the CNTF/PANI-5C increased to 134.7 MPa, with a slightly declined elongation at break (18.4%), which was due to the continuously reinforced CNTF skeleton, the better interface adhesion, and the dense filling of the inner core in the CNTF with the PANI layer. However, the tensile strength of the CNTF/PANI-15C was reduced to 124.5 MPa, meaning that the PANI nano “sea-cucumber-like” structure constituted a loose three-dimensional porous microstructure, with the relatively worse mechanical property.

### 3.2. Electrochemical Performance Analysis

The electrochemical properties of the CNTF/PANI nanocomposite electrodes were characterized by the CV, GCD, and EIS curves. As shown in Figure 8A, the CNTF and the acidified CNTF working electrodes showed the rectangular characteristic CV curves, representing the typical electric double-layer capacitor behavior. The CNTF/PANI electrodes showed the obvious three pairs of peaks attributed to the redox reactions of the PANI, showing the typical pseudocapacitor behavior. The peaks, A_1_/C_1_, are attributed to the leucoemeraldine–emeraldine transition, the peaks, A_2_/C_2_, corresponded to the intermediates of the hydroquinone/benzoquinone during the PANI redox reaction, and the peaks, A_3_/C_3_, represented the redox transition between the emeraldine–pernigraniline states. The CV curve of the CNTF/PANI-15C showed the strongest redox peak, and the smallest potential difference between the A_1_ and C_1_ peaks, indicating the best reversibility of the electrode reaction. The formation of the uniform, ordered, and porous PANI nano “sea-cucumber-like” structure layer in the CNTF/PANI-15C increased the specific surface area of the PANI layer, and effectively expanded the exposed area in the electrolyte. Thus, the quinoid N atoms in the doped PANI macromolecular chains can be easily protonated, and then interacted with the SO_4_^2−^ in the electrolyte to promote the redox reaction and store the charges in the CNTF/PANI-15C nanocomposite electrode. On the other hand, the loose PANI layer on the surface of the CNTF was stacked to form plentiful pores, which encouraged the ions in the electrolyte to arrive at the CNTF core layer quickly and enhanced the charge-storage capacity. Figure 8B shows the CV curves of the CNTF/PANI-15C at different scan rates. Because of the influence of the internal resistance of the nanocomposite electrode with the increase in the scanning rate, the redox peak was gradually widened, and the cathodic and anode peaks gradually shifted to negative and positive values, respectively [60].

The GCD curves of the pure CNTF, acidified CNTF, and CNTF/PANI nanocomposite electrodes, under the constant current density of 1 A/g, are shown in Figure 9A. The GCD curves of the pure CNTF and the acidified CNTF show a standard symmetrical triangle of the typical electric double-layer capacitor behavior. The GCD curves of the CNTF/PANI, prepared by the CVEP method, show a shape similar to a triangle, and some nonstraight areas indicate the existence of a redox reaction process. This typical pseudocapacitor behavior of the CNTF/PANI was consistent with the previous analysis of the CV curves.

With the increase in the electrochemical polymerization cycle period, the discharge times and specific capacitances of the CNTF/PANI-5C, CNTF/PANI-10C, and CNTF/PANI-15C nanocomposite materials were gradually increased, and the CNTF/PANI-15C nanocomposite electrode showed the longest discharge time, indicating the highest specific capacitance and an energy density of 903.6 F/g and 45.2 Wh/kg, respectively. The specific capacitances, energy densities, and power densities of the composite electrodes were calculated by Equations (1)–(3). The power density of the electrode material is 0.3 kW/kg, with a test potential window from 0 to 0.6 V. The properties of this work, and the previously reported results, are compared in Table 2 [19,60,61,62,63,64,65]. The PANI nanowire layer, with a similar structure as the “sea-cucumber-like” layer, was constructed on the CNTF conductive skeleton through electrochemical polymerization. Then, the 3D nano convex structure of the “sea-cucumber-like” layer was grown orderly and stacked to form the loose PANI layer with pores with the increase in the cyclic voltametric polymerization periods. This critical structure can effectively increase the exposed area of the electrode in the electrolyte and can provide enough active sites for ion adsorption. In addition, the conductive skeleton, with crisscrossing CNT fibers and fiber bundles, can act as an internal current collector in the CNTF/PANI. The gaps in these PANI pores helped the electrolyte ions to reach the inner CNTF current collector layer for much more charge storage. The doped PANI nanowires, with the intrinsically preferred conductivity and large aspect ratio, were also conducive to the transmission of electrons on the surface of the PANI layer. All the mentioned factors will play a positive role in suppressing the voltage drop of the supercapacitor. Figure 9B shows the GCD curve of the CNTF/PANI-15C nanocomposite at different current densities. The GCD curves were all close to the triangle shape, confirming the much smaller internal resistance of the CNTF/PANI-15C. The specific capacitance decreased from 903.6 to 668.4 F/g as the current density increased from 1 to 10 A/g. This is because, as the current density increases, the ion diffusion rate in the solution is lower than the electron transfer rate, and the charging and discharging time becomes shorter. It maintained 73.9% of the initial value at the current density of 1 A/g, indicating the excellent rate performance and sustainability of the CNTF/PANI-15C as an electrode material under high current density. 

Figure 9C shows the specific capacitance changes of all the CNTF/PANI under different current densities. The specific capacitances were all decreased with the increase in the current density, and the CNTF/PANI-15C nanocomposite electrode presented the best specific capacitance at the same current density.

The EIS testing was used to evaluate the structure, resistance, and charge transfer capacity between the electrode surface and the electrolyte during the charging and discharging process of the CNTF/PANI nanocomposite electrodes. Figure 9D shows the EIS testing results of the CNTF/PANI-5C, CNTF/PANI-10C, and CNTF/PANI-15C nanocomposite electrodes, and the CNTF/PANI-15C nanocomposite electrode after 2000 cycles of GCD in 1-M H_2_SO_4_ electrolyte, under the AC voltage of 5 mV, and in a frequency range from 0.01 Hz to 100 kHz. All the curves were close to a straight line, with a large slope in the low-frequency region, indicating higher conductivity and low internal impedance [66,67]. This ideal capacitance behavior is mainly due to the pseudocapacitance characteristics provided by the PANI. The different components used in this circuit have physical meanings, which are as follows: The intercept value between the curve in the high-frequency region and the X-axis was the equivalent series resistance (R_S_), the interfacial charge transfer resistance (R_CT_), the Warburg impedance (W), and the constant phase element (CPE) [68]. It can be seen from the enlarged part of Figure 9D that the R_S_ value gradually decreased with the increase in the polymerization cycle period, and that it attained 2.60 Ω of the CNTF/PANI-15C. It can also be found that the semicircle width gradually decreased in the intermediate frequency region with the increase in the polymerization cycle period, and the CNTF/PANI-15C showed the smallest R_CT_ value of 0.59 Ω. The above results further confirm that the microporous pores and the surface 3D nanoprotrusion (sea-cucumber-like) structure of the PANI nanowires can increase the specific surface area and provide enough active sites for the ion adsorption to improve the ion transmission rate when they are used as the working electrodes.

The stabilities of different CNTF/PANI nanocomposite electrodes as electrodes after 2000 GCD cycles and 500 cycles of refolding are shown in Figure 10. It was found that all the CNTF/PANI nanocomposite electrodes can maintain more than 90% of the initial values after 2000 GCD cycles (Figure 10A). This is mainly because that PANI was centered on the axial direction of the acidified CNT fibers and was grown uniformly, with the stronger interface adhesion. The specific capacitance of the CNTF/PANI-15C nanocomposite electrode dropped to 859.4 F/g (95.1% of 903.6 F/g) after 2000 GCD cycles, indicating that the design of the PANI growth morphology by electrochemical polymerization was further beneficial to improving the electrochemical performance of the nanocomposite electrode. Comparing the EIS spectrum of the CNTF/PANI-15C nanocomposite electrode after 2000 cycles of GCD, it is apparent that a greater charge transfer resistance is observed (Figure 9D). These changes are due to the faradaic and nonfaradaic reactions at the interface during the charge/discharge process and the ion transport from the electrolytes into the electrode. The EIS spectrum is in accordance with the results of the 2000 GCD cycles.

As is seen in Figure 10B, the CNTF/PANI-15C nanocomposite electrode can be easily folded, showing excellent flexibility. After folding the CNTF/PANI-15C nanocomposite electrode 50, 100, 250, and 500 times, the stability of the CV and GCD were tested, and the results are shown in Figure 10C,D. There was almost no change in the electrochemical performance after 500 refoldings, which shows the excellent stabilities of the structure and the electrochemical performance.

## 4. Conclusions

In conclusion, the continuously enhanced CNTF was prepared as a flexible conductive framework by the FCCVD method. Furthermore, a series of novel CNTF/PANI nanocomposite electrode materials with different structures were prepared by CVEP, and were characterized by FESEM, TEM, FTIR, Raman, XRD, XPS, and other analysis methods. The growth model, morphology, apparent color, and loading amount of the PANI can be controlled by the different cycles of CVEP. The PANI layer was centered on the axial direction of the CNT fibers and was grown uniformly to completely wrap the inner CNT fiber core in the early stage. The covered PANI layer was further grown to form the PANI nanowires with distinctive “sea-cucumber-like” 3D nanoprotrusion structures. The “sea-cucumber-like” structures were finally stacked into microporous channels to increase the specific surface area, which was beneficial to providing enough active sites for ion adsorption and to achieving rapid ion transport and charge transfer. The CV, GCD, EIS, and long-period cycle tests show that the specific capacitance and energy density of the CNTF/PANI-15C were attained to 903.6 F/g and 45.2 Wh/kg, respectively, at a current density of 1 A/g, and the excellent performance of a 73.9% retention rate at the high current density of 10 A/g. The specific capacitance was maintained at 95.1% of the initial value after 2000 GCD cycles, and it was kept almost unchanged after 500 cycles of bending. In addition, the CNTF/PANI-15C nanocomposite electrode showed a higher tensile strength of 124.5 MPa, with a breaking elongation of 18.9%. Therefore, the CNTF/PANI nanocomposite electrode, with excellent flexibility, mechanical properties, a higher specific capacitance and rate performance, and a stable cycle performance will provide the potential ideal electrode material in the fields of supercapacitors and smart, wearable, and flexible energy-storage devices.

## Figures and Tables

**Figure 1 nanomaterials-12-00008-f001:**
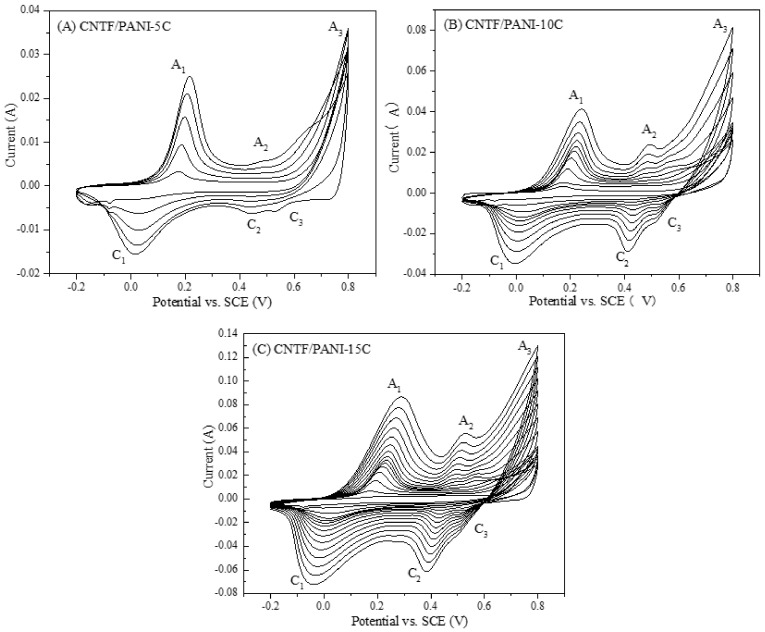
(**A**) Cyclic voltammogram of electrochemical synthesis of the CNTF/PANI-5C sample; (**B**) CNTF/PANI-10C sample; and (**C**) CNTF/PANI-15C sample.

**Figure 2 nanomaterials-12-00008-f002:**
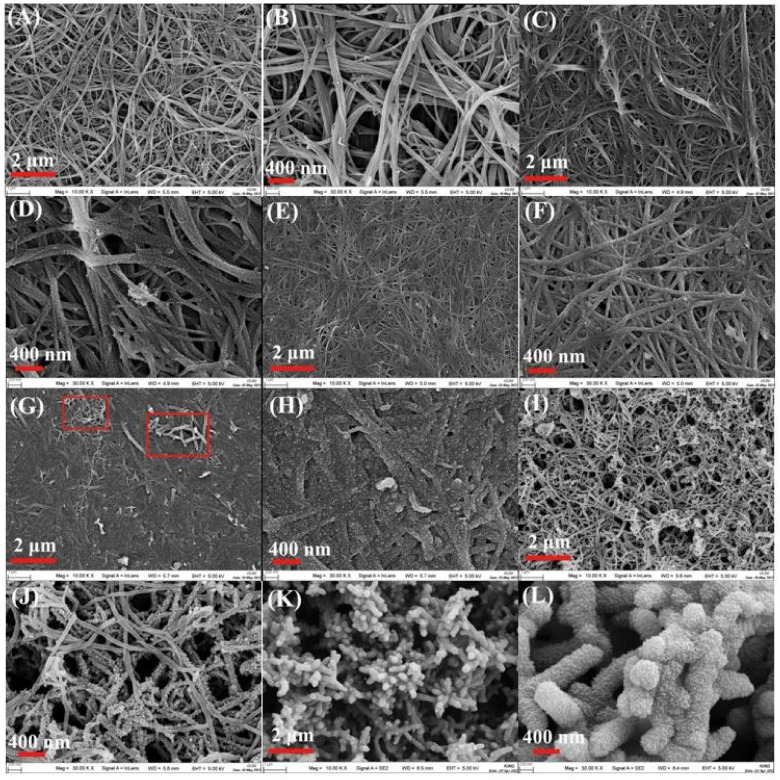
(**A**,**B**) FESEM images of the acidified CNTF; (**C**,**D**) CNTF/PANI-1C nanocomposites; (**E**,**F**) CNTF/PANI-3C nanocomposites; (**G**,**H**) CNTF/PANI-5C nanocomposites; (**I**,**J**) CNTF/PANI-10C nanocomposites; and (**K**,**L**) CNTF/PANI-15C nanocomposites.

**Figure 3 nanomaterials-12-00008-f003:**
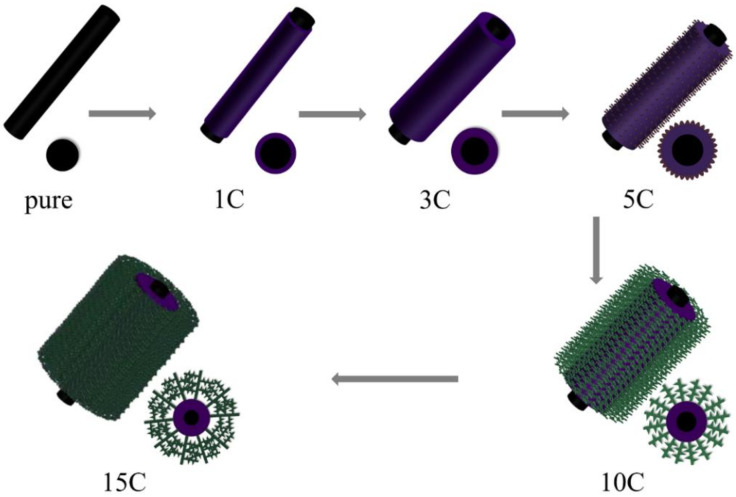
Schematic illustration of the formation processes of PANI on the surface of the CNTF (“C” denotes cycles).

**Figure 4 nanomaterials-12-00008-f004:**
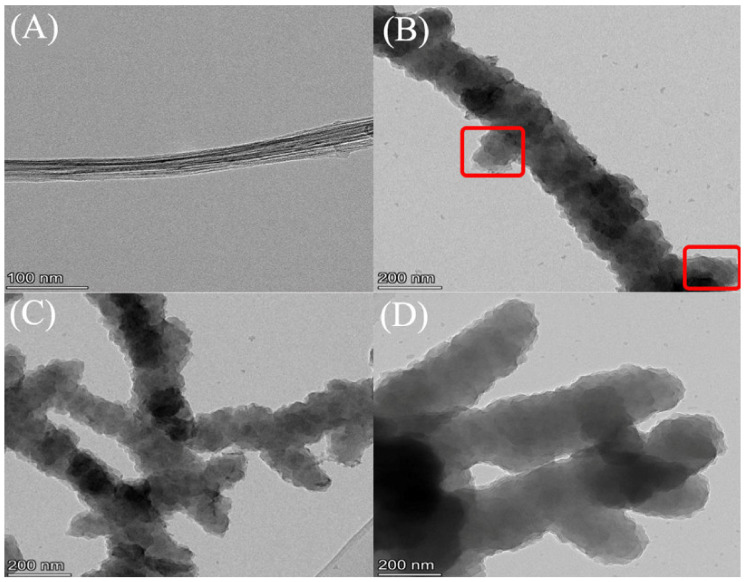
(**A**) TEM images of the acidified CNTF; (**B**) CNTF/PANI-5C; (**C**) CNTF/PANI-10C; and (**D**) CNTF/PANI-15C samples.

**Figure 5 nanomaterials-12-00008-f005:**
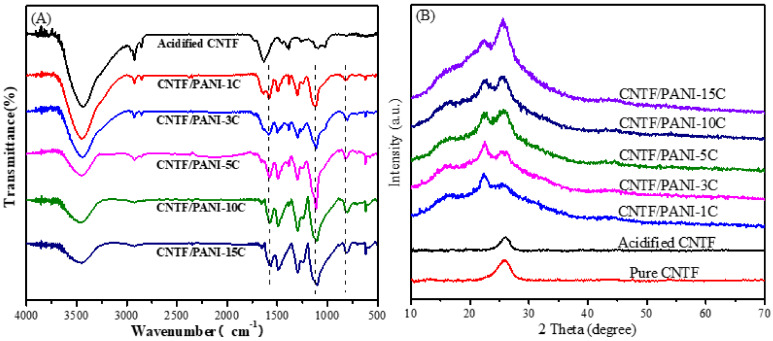
(**A**) FTIR spectra of acidified CNTF, CNTF/PANI-1C nanocomposites, CNTF/PANI-3C nanocomposites, CNTF/PANI-5C nanocomposites, CNTF/PANI-10C nanocomposites, and CNTF/PANI-15C nanocomposites; (**B**) XRDs of pure CNTF, acidified CNTF, CNTF/PANI-1C nanocomposites, CNTF/PANI-3C nanocomposites, CNTF/PANI-5C nanocomposites, CNTF/PANI-10C nanocomposites, and CNTF/PANI-15C nanocomposites.

**Figure 6 nanomaterials-12-00008-f006:**
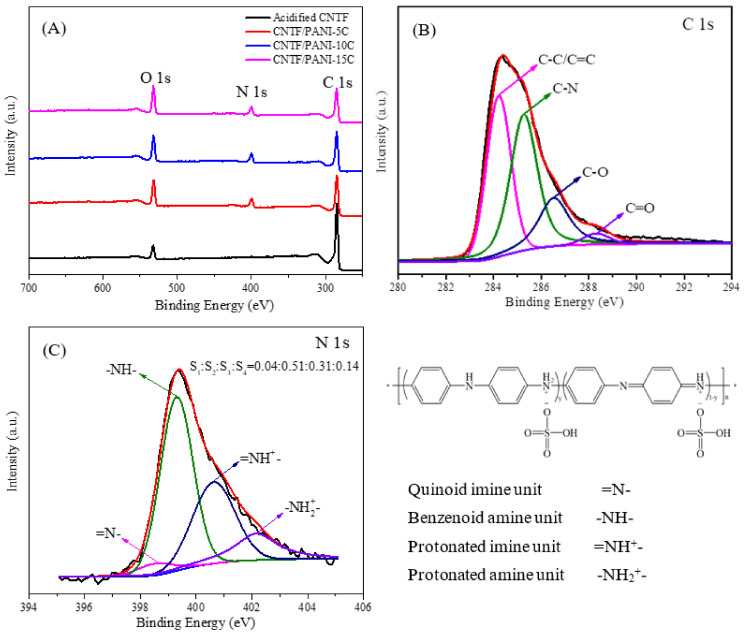
(**A**) X-ray photoelectron spectra of acidified CNTF and CNTF/PANI nanocomposites; (**B)** C 1s spectrum; and (**C**) N 1s spectrum of CNTF/PANI-15C nanocomposites.

**Figure 7 nanomaterials-12-00008-f007:**
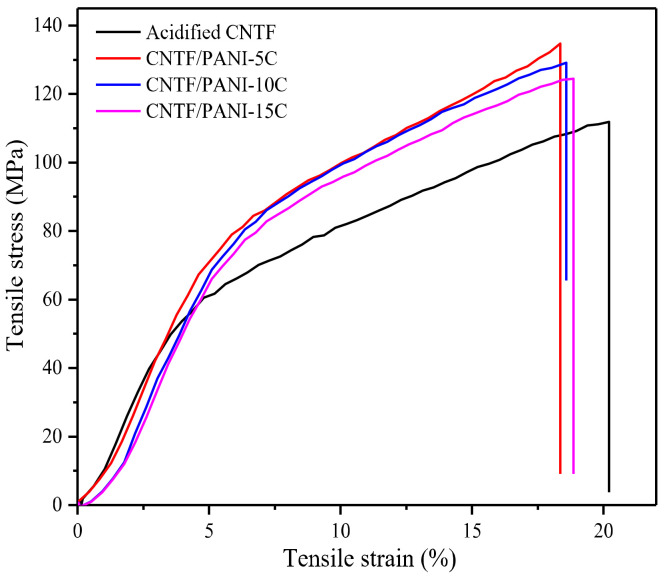
Stress–strain curves of the acidified CNTF, CNTF/PANI−5C, CNTF/PANI-10C, and CNTF/PANI-15C nanocomposites.

**Figure 8 nanomaterials-12-00008-f008:**
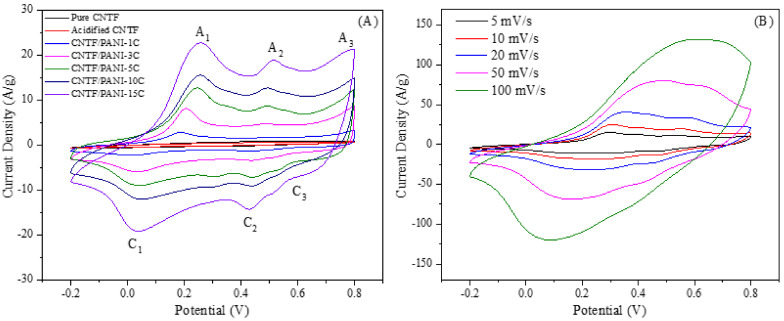
(**A**) CV curves of pure CNTF, acidified CNTF, and CNTF/PANI nanocomposites, measured at a scan rate of 10 mV/s in 1.0-M H_2_SO_4_ solution; and (**B**) CV curves of CNTF/PANI-15C nanocomposites, measured at various scan rates in 1.0-M H_2_SO_4_ solution.

**Figure 9 nanomaterials-12-00008-f009:**
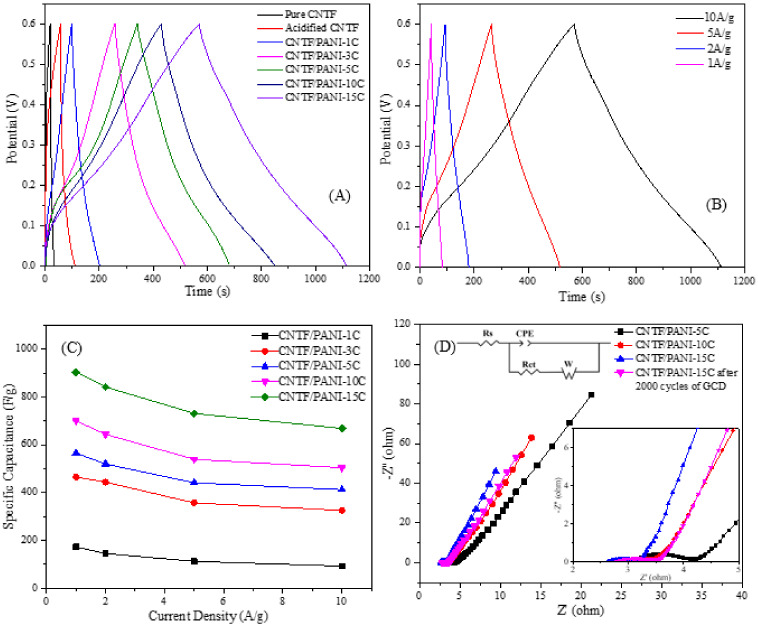
(**A**) GCD curves of pure CNTF, acidified CNTF, and CNTF/PANI nanocomposite electrodes at a current density of 1 A/g in 1.0-M H_2_SO_4_ solution; (**B)** CNTF/PANI-15C at different current densities (1, 2, 5, and 10 A/g); (**C**) The specific capacitances of CNTF/PANI nanocomposites at different current densities; (**D**) Nyquist plots for the CNTF/PANI-5C, CNTF/PANI-10C, and CNTF/PANI-15C nanocomposite electrodes, and the CNTF/PANI-15C nanocomposite electrode, after 2000 cycles of GCD. Inset shows the equivalent circuit for the EIS diagram and the high-frequency region.

**Figure 10 nanomaterials-12-00008-f010:**
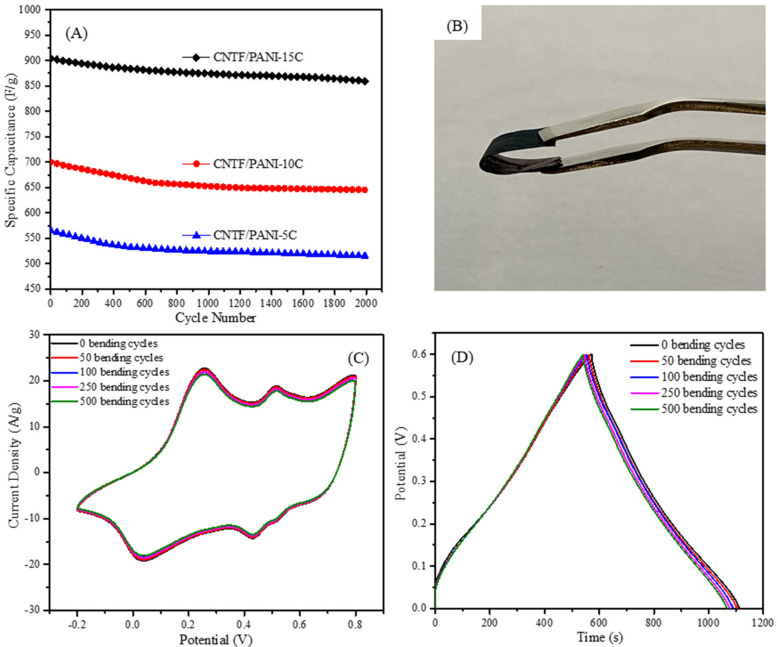
(**A**) The cycling performance of CNTF/PANI-5C, CNTF/PANI-10C and CNTF/PANI-15C nanocomposite electrodes at a current density of 1 A/g in 1.0-M H_2_SO_4_ solution; (**B**) Image of CNTF/PANI-15C nanocomposite electrodes bent to 180°; (**C**) CV curves of CNTF/PANI-15C electrode after different bending cycles; (**D**) GCD curves of CNTF/PANI-15C electrode after different bending cycles.

**Table 1 nanomaterials-12-00008-t001:** The influence of different electrochemical polymerization cycles.

Samples	Cycles (c)	Scan Rate (mV/s)	PANI Content (wt%)	Thickness (μm)	C (1 A/g) (F/g)	E Wh/kg
CNTF	-	-	0	15.0	25.6	1.2
Acidified CNTF	-	-	0	15.0	88.9	4.4
CNTF/PANI-1C	1	10	62.5	22.6	172.3	8.6
CNTF/PANI-3C	3	10	73.2	26.3	464.8	23.2
CNTF/PANI-5C	5	10	121.5	30.7	565.0	28.2
CNTF/PANI-10C	10	10	163.1	51.1	700.1	35.0
CNTF/PANI-15C	15	10	441.1	143.1	903.6	45.2

**Table 2 nanomaterials-12-00008-t002:** Comparison of electrode performance reported in related literature.

Samples	Current Density (A/g)	Electrolyte	Specific Capacitance (F/g)	Cycle Stability	Tensile Strength (MPa)	Ref.
C-CNTs/TC-PANI	1.0	1 M H_2_SO_4_	531.0	92.3% (1000C)	1.75	[19]
RGO/CNs/PANI	1.0	1 M H_2_SO_4_	787.3	92.2% (2000C)	10.1	[60]
rGO/Fe_3_O_4_/PANI	1.0	0.5 M H_3_PO_4_	283.4	78.0% (5000C)	9.2	[61]
CMS@CZL	1.0	6 M KOH	238.0	94.3% (10,000C)	0.08	[62]
PLA/CNTs/PANI	1.0	1 M H_2_SO_4_	510.3	115.0% (2000C)	18.7	[63]
PANI/MWCNTs/PVC	1.0	1 M H_2_SO_4_	801.1	90.6% (2000C)	44.1	[64]
PANI-CC	1.0	1 M H_2_SO_4_	691.0	94.0% (2000C)	-	[65]
CNTF/PANI-15C	1.0	1 M H_2_SO_4_	903.6	95.1% (2000C)	124.5	This work

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
