# Peer review of "Continuously Reinforced Carbon Nanotube Film Sea-Cucumber-like Polyaniline Nanocomposites for Flexible Self-Supporting Energy-Storage Electrode Materials"

_nanomaterials, 2021, doi:10.3390/nano12010008_

Round 1

Reviewer 1 Report

Research in Supercapacitors is evolving these days as it is a significant energy storage device. Various electrode materials including metal oxides, nitrides, carbon, and conducting polymers have been developed for supercapacitor applications. Authors have synthesized CNTF/PANI nanocomposites on film electrodes using a potentiostat deposition such as Electropolymerization and examined these electrodes for supercapacitor applications. The authors have presented a neat characterization with a good energy storage performance and tested for reversibility of 2000 cycles. The composite based on PANI and carbon materials could enhance the capacitance due to the synergetic contribution from the material and mechanical properties. This is a reasonable contribution with good nanomaterials and electrochemistry insights; however, some parts of the manuscript (text and discussion) can be improved. A moderate revision is required before rendering a final decision.

My specific comments are below:

  • What is meant by “continuously reinforced”?
  • The voltage window needs to be mentioned in the abstract, based on the window the capacitance varies.
  • The cucumber-like polyaniline/carbon nanotube-COOH (PANI/CNT-COOH) composite material reported in the literature X. He et al (European Polymer journal 83 (2016) 53-59) need to be discussed and highlight the novelty of the submitted work by the authors.
  • Page 2; “Nevertheless, most of the above composites are not flexible enough.” Is flexibility mean mechanical bending etc. Please be specific.
  • Manickam Minakshi has reported on PANI, and other polymer-related electrodes (including polymerization) please include and discuss in the appropriate sections (1 and 3).
  • Section 2.5 – please provide the conditions for EIS measurements and the equation for power density.
  • Page 5; what is “ANI monomer”? PAIN should read as “PANI”.
  • The peak labels A1, A2, A3, and C1, C2, C3 need to be explained in the discussion along with their redox processes.
  • What are the merits of having a sea cucumber-like structure for supercapacitor applications?
  • Figure 3 – 1C denotes either a charge rate (or) cycles?
  • Ramman should read as “Raman”
  • Is the electrochemical reaction due to proton insertion/de-insertion from H2SO4 electrolyte?
  • Why were the specific capacitances decreased significantly with the increase of current density?
  • Pages 14 – 15 EIS interpretations on higher conductivity and low internal impedance need to be referred to the relevant literature (Dalton Transactions 44 (2015) 20108; and Nanoscale 10 (2018) 13277).
  • Please provide an equivalent circuit for the EIS diagram.
  • The shape of the EIS curves (under different current densities) and the cyclic performance need to be connected and should show synergy.
  • The power density of the electrode material is one of the main factors which limit the electrochemical potential of the material; please discuss the potential window for the materials tested.

Reviewer 2 Report

  1. the scale bars in the images are not clear. Please add.
  2. How did you measure the thickness?
  3. Does all the samples containing polyaniline show cucumber-like structures on the surface of CNT film?
  4. Authors focus the cucumber-like morp[hology very much, I suggest the performance comparison with non-cucumber sample ? and further, discuss the results.
  5. I suggest attaching the video of electrode material to show excellent flexibility. The changes on the flexibility by the acid treatment should be compared with the original/nonacid treated sample.
  6. What the excellent results are attributed to? need to discuss.
  7. The authors stated that acid treatment brings the electrode material more flexible and self supporting? need elaborate with suitable ref in introduction last paragraph and other relevant parts.
  8. IR and XRD graphs can be given in Figure a and b of the same number.
  9. The authors missed the important relevant literatures to review. I suggest to reviewing these. Journal of Colloid and Interface Science 600, 740-751; Electrochem 2 (2), 236-250

Round 2

Reviewer 1 Report

Dear Authors,

This reviewer has read both the author's responses to my queries raised earlier and as well the highlighted (colored) part of the revised manuscript. The current version of the work is comprehensive and reads well. In this reviewer's opinion, the revised manuscript should be suitable to accept as-is and publish. 

Reviewer 2 Report

The authors revised the manuscript as per the reviewer's suggestion therefore I recommend to accept.